# Dynamic Changes in Microvascular Density Can Predict Viable and Non-Viable Areas in High-Risk Neuroblastoma

**DOI:** 10.3390/cancers15030917

**Published:** 2023-02-01

**Authors:** Laura Privitera, Layla Musleh, Irene Paraboschi, Olumide Ogunlade, Olumide Ogunbiyi, J. Ciaran Hutchinson, Neil Sebire, Paul Beard, Stefano Giuliani

**Affiliations:** 1Wellcome/EPSRC Centre for Interventional and Surgical Sciences, University College London, London W1W 7TY, UK; 2Cancer Section, Developmental Biology and Cancer Programme, UCL Great Ormond Street Institute of Child Health, London WC1N 1EH, UK; 3Department of Pediatric Surgery, San Camillo-Forlanini Hospital, 00152 Rome, Italy; 4Department of Paediatric Urology, Fondazione IRCCS Ca’ Granda Ospedale Maggiore Policlinico di Milano, 20122 Milan, Italy; 5Department of Medical Physics and Biomedical Engineering, University College London, London WC1N 1EH, UK; 6Department of Histopathology, Great Ormond Street Hospital for Children NHS Trust, London WC1N 3JH, UK; 7Department of Specialist Neonatal and Paediatric Surgery, Great Ormond Street Hospital for Children NHS Trust, London WC1N 3JH, UK

**Keywords:** high-risk neuroblastoma, microvascular density, optical imaging, photoacoustic imaging, image-guided surgery

## Abstract

**Simple Summary:**

Neuroblastoma (NB) is the most common extracranial tumor in children. Despite the development of new therapeutic options, high-risk patients still have poor long-term survival. Distinguishing viable malignant tissue from necrotic areas and surrounding healthy tissue is pivotal during surgery. Variations in the tumor vascular architecture could guide surgeons to remove only viable tumor regions, achieving a safer and more efficient resection. In this paper, using histopathology scanned slides, we first examined variations in the vasculature density of different tumor regions, showing that viable and necrotic areas are associated with a distinct vascular signature. Then, we scanned an excised human NB specimen using a custom-made high-definition preclinical imaging device based on photoacoustic imaging, which is ideal for imaging vasculature, showing that the acquired images can differentiate macroscopic regions within the tumor.

**Abstract:**

Despite aggressive treatments, the prognosis of high-risk NB remains poor. Surgical oncology needs innovative intraoperative devices to help surgeons discriminate malignant tissue from necrotic and surrounding healthy tissues. Changes within the tumor vasculature could be used intraoperatively as a diagnostic tool to guide surgical resection. Here, we retrospectively analyzed the mean vascular density (MVD) of different NB subtypes at diagnosis and after induction chemotherapy using scanned histological samples. One patient was prospectively enrolled, and an ex vivo photoacoustic imaging (PAI) scan was performed on two representative sections to assess its capacity to discriminate different tumor regions. We found that post-chemotherapy, viable areas of differentiating NBs and ganglioneuroblastomas are associated with higher MVD compared to poorly differentiated NBs. Early necrotic regions showed higher MVD than late necrotic and viable regions. Finally, calcified areas showed significantly lower MVD than any other histological component. The acquired PAI images showed a good high-resolution ex vivo 3D delineation of NB margins. Overall, these results suggest that a high-definition preclinical imaging device such as PAI could potentially be exploited to guide surgical resection by identifying different vasculature signatures.

## 1. Introduction

Neuroblastoma (NB) is the most common extracranial solid tumor in childhood, accounting for 8–10% of all pediatric cancers and about 15% of all pediatric tumor-related deaths [1]. Treatment for NB is decided according to the International Neuroblastoma Risk Group (INRG) stratification, which combines the following criteria: INRG stage, age, histology, tumor differentiation grade, MYCN amplification status, presence of chromosomal aberration, and tumor cell ploidy [2]. Based on this classification, high-risk NB patients account for approximately 70% of the total cases, with a long-term survival rate of less than 50% [3]. The clinical management of high-risk patients remains challenging and involves an intensive multimodal treatment with induction chemotherapy, surgery, high-dose chemotherapy, stem cell transplant, immunotherapy, and radiation [3]. 

Several studies have shown the importance of microvascular changes in the prognosis of NB [4,5,6,7,8]. The anatomy and characteristics of blood vessels in the case of NB with unfavorable INRG prognostic features are unique, with large, branched, and deformed structures compared to tumors with favorable clinical and biological characteristics [8]. However, distinctive microvascular traits have not yet been investigated as potentially valuable diagnostic features to differentiate histopathological categories and tumor response to induction chemotherapy. NBs are also characterized by important histological changes after treatment, with areas of necrosis, calcification, fibrosis, and cancer-related inflammation [9,10,11]. Pathologists commonly report all these chemotherapy-induced modifications as “non-viable areas of NB”.

Due to the high risk of surgical complications, it is crucial to know intraoperatively whether any residual tissue in the tumor bed is viable, as it might reduce morbidity and mortality associated with surgery [12,13]. Radical excision of NB carries the potential risk of either permanent injury to vital organs (i.e., kidney and liver) or severe bleeding due to the tumor’s natural propensity to encase main retroperitoneal vessels [14]. At present, oncology surgeons have no tool to differentiate viable versus calcified, fibrotic, or necrotic areas within the tumor. Therefore, the surgeon has to perform an extensive resection in order to remove more than 95% of the mass [15,16]. 

Our hypothesis is that differences in the mean vascular density (MVD) of viable and non-viable areas might be used intraoperatively to guide surgical excision of the tumor. In particular, we believe that photoacoustic imaging (PAI), a promising non-invasive high-definition optical imaging technique, could be used intraoperatively to visualize and define differences in the microvasculature of viable and non-viable parts of NB [17,18,19]. PAI is based on the absorption of laser light of specific wavelengths (often in the near-infrared spectrum) by intrinsic components of the imaged tissue. By imaging at multiple wavelengths, the distribution of hemoglobin, lipid, and water can be mapped with high resolution [20,21]. 

Therefore, the aim of this study is first to retrospectively review pathological specimens of high-risk NB patients to evaluate differences in the MVD of viable and non-viable areas. Finally, using a pre-clinical platform, we assessed the ex vivo abilities of PAI to acquire images of excised NB human samples. 

## 2. Materials and Methods

### 2.1. Patient Selection

For the first part of the study, we retrospectively analyzed histological samples of 30 patients with high-risk NB treated at Great Ormond Street Hospital (GOSH) (London, UK) from January 2013 to December 2019. Patients were selected using the following inclusion criteria: high-risk NB (based on the INRG stage at diagnosis); histology-based diagnosis of NB (according to the International Neuroblastoma Pathology Classification); patients younger than 16 years old at the time of diagnosis; histopathology blocks available both at diagnosis and after surgical excision. 

The following histological subtypes were included in the study: undifferentiated NB (uNB, *n* = 10), poorly differentiated NB (pdNB, *n* = 10), and differentiating NB (dNB, *n* = 10). All patients underwent induction chemotherapy before surgical excision, which consisted of either the COJEC (rapid Cisplatin, Vincristine, Carboplatin, Etoposide, and Cyclophosphamide) or the modified N7 regimen. This was followed by Myeloablative Busulfan/Melphalan (BuMel) consolidation in case of residual disease, as per the International Society of Pediatric Oncology (SIOP) protocol. 

A high-risk NB patient was prospectively enrolled to perform an ex-vivo PAI scan on representative sections of the resected tumor. Clinical data were retrieved using GOSH IT databases (EPIC and OMNI). 

### 2.2. Samples Processing

The retrospective part of the study involved the use of formalin-fixed, paraffin-embedded tissues. For each patient, hematoxylin and eosin (H&E) slides were retrieved from the laboratory information management system of the histopathology department at GOSH. Samples were obtained after needle biopsies at diagnosis and from surgical resection after induction chemotherapy. The sections of nine patients (udNB = 3, pdNB = 3, dNB = 3) were also stained with immunohistochemistry for CD31 (pan-endothelial marker) and used as a positive control for image processing. H&E and CD31 stained slides were digitized at 40× magnification using a ScanScope XT scanner (Aperio Technologies, Vista, CA, USA). The scanned slides were anonymized and saved on a secure external hard drive.

For the human pilot study, surgically excised material underwent routine histopathological processing. The remaining formalin-fixed discarded material was anonymized and stored at the histopathology department to be subsequently imaged with the PAI scanner. 

### 2.3. Image Processing of the H&E and CD31 Pathology Slides

The image analysis was performed manually using an open-source software platform for whole slide image analysis (QuPath) [22]. Briefly, a grid consisting of regular lines with identical lengths and distances was overlaid by the QuPath software on the image (square area ≈ 546,494 µm^2^). Blood vessels with a diameter between 50 µm and 150 µm were manually labeled in five 3 × 3 squared regions of interest (ROIs). In each region, the MVD was defined as the ratio of the vascular tissue area to the total tissue area in the field of interest. 

The representative number of ROIs was established using a Systematic Uniform Random Sampling (SURS) of 11 ROIs. The first ROI was selected randomly, while the remaining regions were taken at regular sampling intervals throughout the tissue. Three consecutive MVD values (included within the 95% confidence interval of the mean standard error) were obtained after the 5th ROI [23,24,25] (Appendix A). 

Hence, five ROIs for each patient were selected from sections obtained at diagnosis. The scanned slides of the samples obtained after the surgical resection were divided into viable areas (5 ROIs), early necrotic areas (5 ROIs), late necrotic areas (5 ROIs), and calcified areas (5 ROIs) (a total of 20 ROIs per patient after induction chemotherapy). Patients MVDs were calculated as the average MVD of the 5 ROIs. Non-viable regions were divided into early necrosis, late necrosis (fibrotic/scarred tissue), and calcified areas to avoid statistical noise, as they have morphologically well-recognizable, distinct characteristics. 

To demonstrate that the rate of vessel detection on the H&E was reliable and reproducible, a further analysis was performed on the CD31 stained slides. Following the same protocol, the MVD of the CD31 stained slides was calculated on approximately the same ROIs of the corresponding H&E and values were compared. 

### 2.4. Data Analysis

MVD values were, imported in an Excel file (MS Excel. Software Office Version 2212) and sorted by patient and histological subtype. After an initial descriptive analysis, we evaluated differences in the MVD of viable areas of different histological subgroups at diagnosis and post-induction chemotherapy. We then analyzed differences in the MVD of viable, early necrotic, late necrotic, and calcified regions post-induction chemotherapy. This analysis was also performed by sorting patients according to their histological subtype after treatment. The non-parametric Kruskal–Wallis test was used to determine significant differences in MVD. H&E versus CD31 detection rate was compared using the Mann–Whitney test for independent data. A *p*-value of less than 0.05 was considered as statistically significant. All analyses were performed using GraphPad Prism version 9.4.0 for Windows (GraphPad Software, San Diego, CA, USA). 

### 2.5. Photoacoustic Imaging

PAI of 2 ex-vivo sections obtained from an excised sample of NB were acquired using a high-resolution planar Fabry–Perot (FP) scanner previously described [26,27,28,29]. The scanner provides a lateral field of view of 20 mm × 20 mm, with a depth-dependent resolution between 50–150 µm and an imaging depth of ~10 mm depending on excitation wavelength [26,27,28,29]. In this study, two different excitation wavelengths were used; 620 nm for higher contrast and lower imaging depth and 920 nm for lower contrast and higher imaging depth. Before imaging, the two formalin-fixed NB samples were washed overnight using deionized water. Acoustic coupling between the samples and the scanner was achieved by inserting a small amount of ultrasound coupling gel.

## 3. Results

At diagnosis, the same number of patients was selected for each of the different histological subtypes (uNB = 10; pdNB = 10; dNB = 10). However, histological assessment after induction chemotherapy showed a very heterogeneous response (Appendix A). In detail, patients in the differentiating group mostly confirmed their histological subtype, with only three cases switching toward more undifferentiated forms and one patient differentiating into ganglioneuroblastoma (GNB) phenotype. Patients belonging to the poorly differentiated subtype had a more mixed response, with 50% switching towards more differentiating phenotypes (*n* = 3 dNB; *n* = 2 GNB) and one patient changing toward an undifferentiated subtype. None of the patients initially selected as uNB confirmed their histology after induction chemotherapy. Three of them showed a complete response to chemotherapy, while the remaining patients were classified as dNB (*n* = 2), pdNB (*n* = 4), and GNB (*n* = 1). Overall, four patients had a complete response, with tumors entirely non-viable after induction chemotherapy (Table 1). 

### 3.1. H&E Staining Is Associated with Reliable Vessels Detection 

To evaluate whether the manual labeling on the H&E slides was a reliable and accurate approach to measure MVD, slides stained with CD31 were manually labeled using approximately the same ROIs, and used as a positive control. A total of 150 ROIs were manually labeled on the H&E slides, and 149 ROIs were selected on the corresponding CD31 stained slides. No significant difference was found in the MVD of the two groups (*p* = 0.50, Mann–Whitney test), confirming that the manual labeling on the H&E staining allows a reliable estimate of the MVD (Figure 1).

### 3.2. Viable Regions of Different Histological Subtypes Show Comparable MVD before and after Chemotherapy

The MVD of viable regions was a significant source of variance among different patients, and within the same patient, before and after treatment (*p* < 0.01, mixed-effect analysis) (Appendix A). No significant difference was found in the MVD of the viable areas of different histological subtypes at the time of diagnosis (Figure 2a–c). After treatment, a lower MVD was observed in dNB (MVD = 0.006 ± 0.004) and GNB tumors (MVD = 0.005 ± 0.003) compared to the pdNB group (MVD = 0.02 ± 0.02) (Figure 2d–f).

### 3.3. MVD Allows the Identification of Different Histological Components within the Excised Tumor after Induction Chemotherapy

Overall, calcified regions had significantly lower MVD (MVD = 0.002 ± 0.001) than viable (MVD = 0.01 ± 0.01), early necrotic (MVD = 0.02 ± 0.01), and late necrotic (M = 0.007 ± 0.006) regions (*p* < 0.0001, *p* < 0.0001, *p* < 0.001, respectively; Kruskal–Wallis test). When comparing necrotic areas within the tumor, early necrotic regions showed significantly higher MVD than late necrotic ones (*p* < 0.05; Kruskal–Wallis test). Although this did not reach significance, early necrotic areas were also associated with higher MVD compared to viable tumor regions (Figure 3a–c). These results were also confirmed after sorting patients based on the different histologies after induction chemotherapy, with the poorly differentiated group showing a significant difference in the MVD of viable areas and late necrotic regions, with higher vessels densities for the former (*p* = 0.03, Kruskal–Wallis test) (Appendix A). 

### 3.4. Photoacoustic Imaging Allows the Detection of Different Regions within the Primary Tumor: Preliminary Ex Vivo Data 

To explore the feasibility of using PAI to distinguish different histological components within the tumor and guide surgical excision, two representative formalin-fixed human surgical sections of a treated high-risk NB tumor were imaged ex-vivo using a pre-clinical PAI system (Figure 4). Macroscopically, the first section showed fairly large areas of viable NB (Figure 4a). Conversely, the second section showed a necrotic core encased in surrounding fibrotic tissue (Figure 4b). The excitation wavelengths used for the PAI scans for each section were chosen to optimize the trade-off between contrast and penetration depth. PAI images of the first section were acquired using an excitation wavelength of 920 nm, which penetrates deeply, enabling the clear demarcation of a nodule of viable tumor (Figure 4a). The 3D nature of the blood vessels within the nodule can be visualized by viewing the fly-through movie in the Appendix A. The PA image of the second section, acquired using an excitation wavelength of 620 nm, allowed a clear distinction between the necrotic core of the tumor and the surrounding fibrotic capsule with a penetration depth of around 3.5 mm (Figure 4b). 

## 4. Discussion

Although neoadjuvant treatments have proven to shrink advanced primitive tumors, the strict encasement of the main abdominal vessels and vital structures makes NB surgical resection challenging [14]. The development of new intra-operative devices that can guide the surgeon during resection is of crucial importance. In this study, we show the possibility of using vascular density to distinguish between viable and non-viable tumor regions. In addition, to the best of our knowledge, we are the first to use PAI on an ex-vivo NB human sample and to show that it can differentiate NB tissue components. This suggests the technology has potential as an intra-operative imaging tool for distinguishing viable vs. non-viable areas of the residual mass. 

In detail, our analysis showed that after induction chemotherapy, calcified tumor areas were significantly less vascularized than any other histological component within the tumor. Similarly, a significant difference was found in the MDV of early and late necrotic areas, with early necrotic areas having higher vessel density. Interestingly, early necrotic areas showed comparable, if not higher, MVD than viable regions, which could be partially explained by the pro-angiogenetic stimuli of the inflammation process present during the early phases of necrosis. This data could be of considerable importance for understanding the significance of the heterogeneous aspect that these tumors have on contrast imaging, the different responses to the pre-operative treatment, and, ultimately, it could be of practical use during surgical resection. 

The importance of tumor microvascularity has been extensively assessed due to its dual nature as prognostic indicator and treatment guidance in several adult malignancies. It has been previously demonstrated that higher tumor vascularity correlates with worse outcomes in malignancies such as breast, prostate, and ovary, representing the substrate for tumor maintenance, growth, and metastatic ability [8,30,31]. An abnormal vascular net is also the rationale for anti-angiogenic therapies, which now increase survival and disease-free lapses in adult patients affected by these malignancies [32]. 

From a quantitative perspective, the vascular architecture of chemo-treated high-risk NB has been previously described by a German study published in 2012. The authors analyzed tumor vessel density in seven patients affected by high-risk NB before and after chemotherapy. In accordance with our findings, samples obtained from early necrotic regions were associated with higher microvascular density [33]. The same group also hypothesized that changes associated with chemo-resistance development included an enhanced pro-angiogenic activity, demonstrating, both in-vitro and in-vivo, a shift to a pro-angiogenic phenotype in chemo-resistant tumors [34]. Effects of neoangiogenic phenomena on NB have been extensively examined in the great effort to better understand the genetics, histology, and biology behind the aggressive behavior of certain tumors and to selectively tailor future therapies. It has been described that more aggressive risk groups display specific abnormal characteristics [35,36,37]. More recently, the description of a disorganized glomeruloid microvascular proliferation in poor outcome NB has been correlated to the hypothesis that angiogenesis could be regulated differently in Schwannian stroma-rich with a better prognosis than in stroma-poor NBs, which are associated with poor outcomes [38]. 

Following this research trend, Tadeo et al. showed that the vascular characteristics of the different neuroblastic tumors correlate to clinical variables. They ultimately concluded that unfavorable histology groups presented a higher blood vessel density with larger, rounder, more deformed, and branching blood vessels and that unfavorable-histology NBs had irregular sinusoid vessels compared to favorable NB, which presented mainly capillaries. Moreover, they found that branched post-capillaries and metarterioles were typically found in unfavorable NB subtypes [8]. Although our results did not show a significant difference in the MVD of different histological subtypes at diagnosis, post-induction chemotherapy, we also found that dNBs and GNBs were less vascularized than pdNBs. 

Surgery plays a pivotal role in the multimodal treatment of NB [39,40]. At the current state of the art, pediatric oncology surgeons exclusively rely on visual inspection, palpation, and personal experience to identify tumor location and extension. Our results show a distinct microvascular signature between viable and non-viable areas. We believe that this difference, combined with the ability of PAI to provide a non-invasive high-resolution 3D imaging of vascular morphology, will inform surgeons during tumor resection, providing an important innovation in the field. In detail, the presence of live imaging at the time of resection could guide the surgeon to better and selectively confirm the presence of viable tumor in difficult anatomic areas, reducing morbidity and mortality. 

While imaging techniques, such as magnetic resonance imaging and computed tomography, often necessitate the use of an exogenous contrast agent to achieve high-resolution images of the vasculature, PAI relies on the strong light absorption of hemoglobin which enables the excellent definition of the microvasculature without employing exogenous contrast [41]. Previous studies have focused on the applicability of PAI for the evaluation of the vascular morphology of solid tumors to either assess pharmacodynamic response to drug treatment or detect cancer based on intrinsic optical absorption contrast [41,42]. Our pilot data also confirm the possibility for PAI to clearly demarcate nodules of viable tumor and distinguish regions with different responses to chemotherapy. Future work will focus on the ex-vivo acquisition of high-resolution 3D anatomical images of tumor vasculature, both in formalin-fixed and fresh tissue, using our preclinical Fabry-Perot PAI system. Ex-vivo validation of PAI’s ability to distinguish viable versus non-viable tissue, with the use of histopathology as a gold standard, will be of crucial importance before translating this novel technology into the clinical environment. 

The main limitation of this study is represented by the manual ROI sampling and blood vessel counting, a labor-intensive and time-consuming process due to the amount of effort required to find valid ROIs on scanned slides. For the clinical translatability of our results, more objective and reproducible methods for microvessel counting could be applied, along with standardization of criteria for evaluating intra-tumor microvascular density and identifying neovascular hot spots within each tumor. Complementary analytical tools, such as artificial intelligence (AI), could be adopted to improve surgeons’ decision-making ability to identify viable tumor areas that need to be radically removed. To define the sensitivity and specificity of our PAI system, future studies will focus on scanning more samples of human NB at different wavelengths. 

## 5. Conclusions

The MVD can be adopted as a reliable marker for distinguishing viable and non-viable areas within chemo-treated tumors. This finding, combined with the possibility of using high-resolution optical devices such as PAI, opens exciting new opportunities for the intraoperative management of NB.

## Figures and Tables

**Figure 1 cancers-15-00917-f001:**
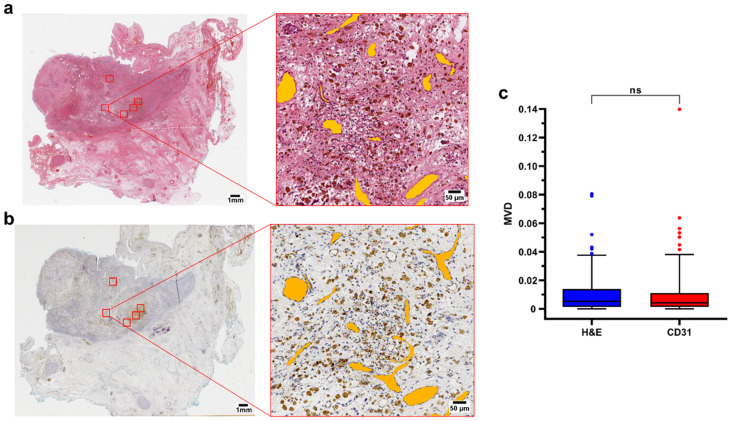
(**a**,**b**) Representative H&E and CD31 immunohistochemistry of a surgical sample (case 2, pdNB). ROIs are highlighted in red in the whole slide section (left). Filled annotations (orange) of the vessels of one of the regions are shown on the right side of the panel; (**c**) Boxplot showing data distribution for H&E (*n* = 150) and CD31 (*n* = 149) ROIs. Box represents the interquartile range. Bars represent the 5–95% range. Abbreviations: MVD = mean vascular density; H&E = hematoxylin and eosin; ns=not significant.

**Figure 2 cancers-15-00917-f002:**
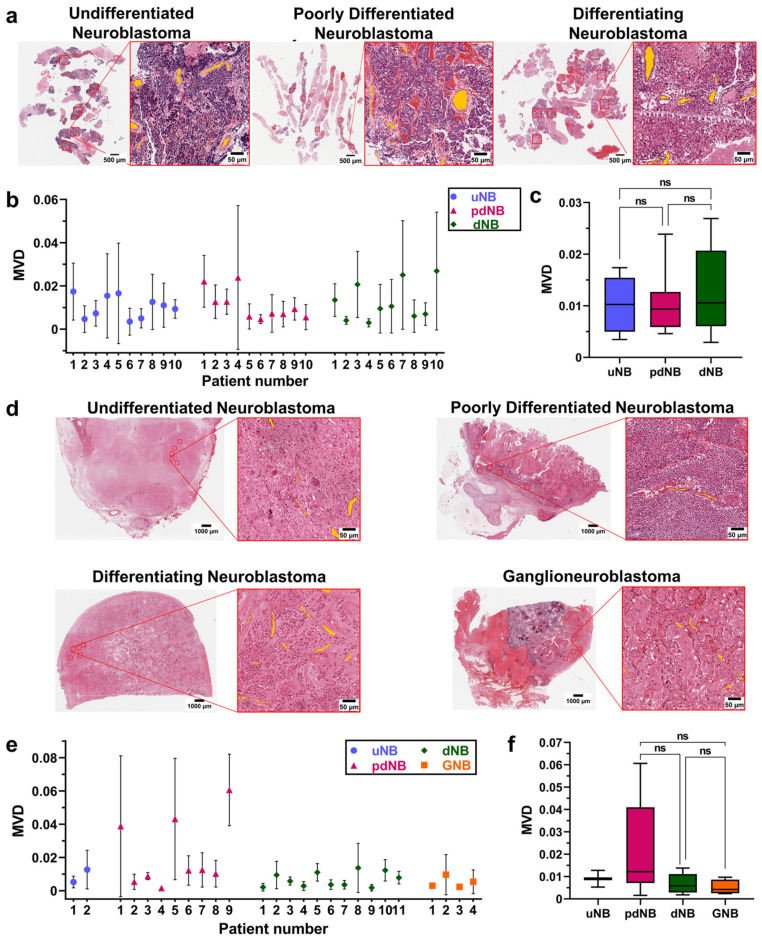
(**a**) Representative H&E sections of the core needle biopsies at the time of diagnosis. Images show the whole slides and the magnification of one of the ROIs for the udNB (left), pdNB (center), and dNB (right) subtype; (**b**) Bars showing mean and standard deviations of the MVD for the different histological subtypes at the time of diagnosis (udNB = 10, pdNB = 10, dNB = 10). (**c**) Boxplot showing the distribution of patients’ MVD grouped by histological subtype at the time of diagnosis (udNB = 10, pdNB = 10, dNB = 10). Box represents the interquartile range. Bars represent the 5–95% range; (**c**) Boxplot showing the distribution of patients’ MVD grouped by histological subtype after induction-chemotherapy (udNB = 2, pdNB = 9, dNB = 11, GNB = 4). Box represents the interquartile range. Bars represent the 5–95% range; (**d**) Representative H&E of the surgical samples after resection. Images show the whole slides and the magnification of one of the ROIs for the udNB (top left), pdNB (top right), dNB (bottom left), and ganglioneuroblastoma (bottom right) subtype; (**e**) Bars showing mean and standard deviations of the MVD for the different histological subtypes after induction chemotherapy (udNB = 2, pdNB = 9, dNB = 11, GNB = 4); (**f**) Boxplot showing the distribution of patients’ MVD grouped by histological subtype after induction chemotherapy (udNB = 2, pdNB = 9, dNB = 11, GNB= 4). Box represents the interquartile range. Bars represent the 5–95% range. Abbreviations: MVD = mean vascular density; uNB = undifferentiated neuroblastoma; pdNB = poorly differentiated neuroblastoma; dNB = differentiating neuroblastoma; GNB = ganglioneuroblastoma; ns = not significant.

**Figure 3 cancers-15-00917-f003:**
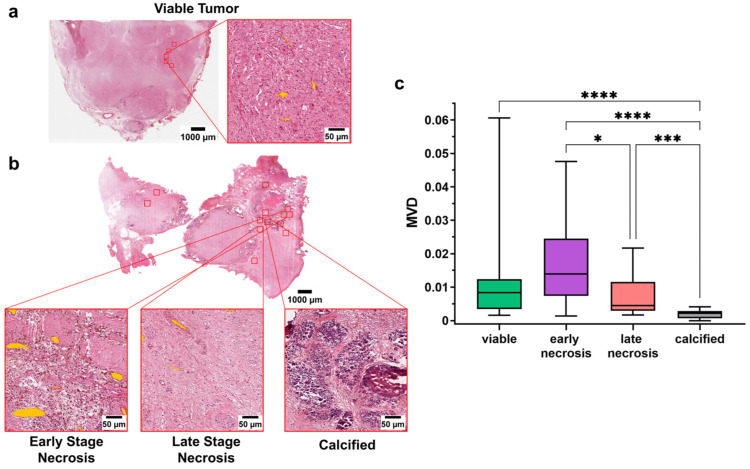
(**a**) Whole slide image and magnification of one of the ROI from a representative patient after induction chemotherapy (udNB). Annotated vessels are shown in orange; (**b**) Whole slide image and magnification of one of the ROI from a representative patient after induction chemotherapy (udNB). Magnified regions show the different histological components of the non-viable components after induction chemotherapy. Annotated vessels are shown in orange; (**c**) Boxplot showing the distribution of patients’ MVD in viable and non-viable regions after induction chemotherapy (viable = 26 patients; early necrosis = 18 patients; late necrosis = 24 patients; calcified = 23 patients). Box represents the interquartile range. Bars represent the 5–95% range. * *p* < 0.05; *** *p* < 0.001; **** *p* < 0.0001.

**Figure 4 cancers-15-00917-f004:**
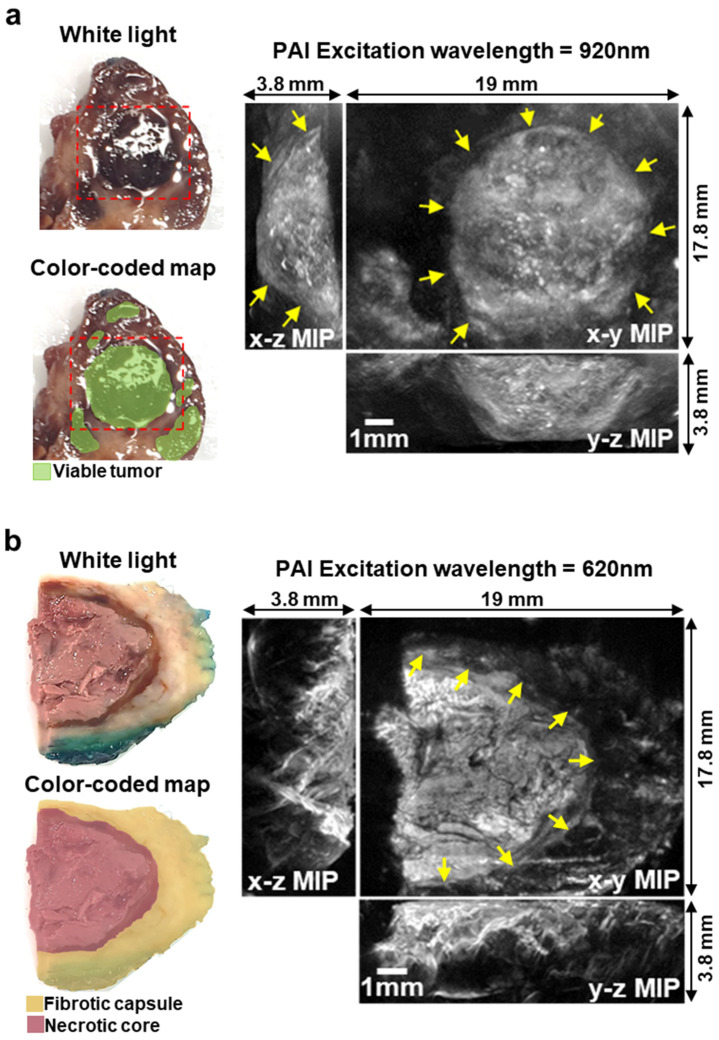
(**a**) White light image and color-coded map of a formalin-fixed surgical section of High-Risk NB after induction chemotherapy. Dashed red line shows the imaged area. On the right, PAI images of the sample (excitation wavelength = 920 nm). The 3D image is shown as maximum intensity projection (MIP). Arrows show the boundary of viable tumor within the PA image. (**b**) White light image and color-coded map of a formalin-fixed surgical section of High-Risk Neuroblastoma after induction chemotherapy. Tumor margins were inked in green for clinical assessment from the histopathology department. On the right, PAI images of the sample (excitation wavelength = 620 nm). The 3D image is shown as MIP. Arrows show the boundary of the fibrotic capsule surrounding the necrotic core of the section.

**Table 1 cancers-15-00917-t001:** Summary of histological subtypes at diagnosis (rows) and after induction chemotherapy (columns) for the entire cohort included in the study (*n* = 30). Abbreviations: uNB = undifferentiated neuroblastoma; pdNB = poorly differentiated neuroblastoma; dNB = differentiating neuroblastoma; GNB = ganglioneuroblastoma; nvNB = non-viable neuroblastoma.

		Histology after Induction Chemotherapy					
		dNB	pdNB	uNB	GNB	nvNB	Total
**Histology at Diagnosis**	**dNB**	6	2	1	1	-	10
	**pdNB**	3	3	1	2	1	10
	**uNB**	2	4	-	1	3	10
	**Total**	11	9	2	4	4	30

## Data Availability

The raw data generated in this study are available upon request from the corresponding author.

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
