# Peer review of "Dynamic Changes in Microvascular Density Can Predict Viable and Non-Viable Areas in High-Risk Neuroblastoma"

_cancers, 2023, doi:10.3390/cancers15030917_

Round 1

Reviewer 1 Report

Intersting concept, however the claim that this will influence the surgical resection needs to be extensively discussed or should be taken out.

It is unclear what the corralation between ex vivo and in vivo use of this technique is. For this to work this will need explaining.

Author Response

We would like to thank you for the valuable feedback. We have updated and expanded the discussion in the manuscript:

“Surgery plays a pivotal role in the multimodal treatment of NB [39,40]. At the current state of the art, pediatric oncology surgeons exclusively rely on visual inspection, palpation, and personal experience to identify tumor location and extension. Our results show a distinct microvascular signature between viable and non-viable areas. We believe that this difference, combined with the ability of PAI to provide a non-invasive high-resolution 3D imaging of vascular morphology, will inform surgeons during tumor resection, providing an important innovation in the field. In detail, the presence of live imaging at the time of resection could guide the surgeon to better and selectively confirm the presence of viable tumor in difficult anatomic areas, reducing morbidity and mortality.

While imaging techniques, like magnetic resonance imaging and computed tomography, often necessitate the use of an exogenous contrast agent to achieve high-resolution images of the vasculature, PAI relies on the strong light absorption of hemoglobin which enables the excellent definition of the microvasculature without employing exogenous contrast [41]. Previous studies have focused on the applicability of PAI for the evaluation of the vascular morphology of solid tumors to either assess pharmacodynamic response to drug treatment or detect cancer based on intrinsic optical absorption contrast [41, 42]. Our pilot data also confirm the possibility for PAI to clearly demarcate nodules of viable tumor and distinguish regions with different responses to chemotherapy. Future work will focus on the ex-vivo acquisition of high-resolution 3D anatomical images of tumor vasculature, both in formalin-fixed and fresh tissue, using our preclinical Fabry-Perot PAI system. Ex-vivo validation of PAI’s ability to distinguish viable versus non-viable tissue, with the use of histopathology as a gold standard, will be of crucial importance before translating this novel technology into the clinical environment.”

Reviewer 2 Report

The authors examined variations in the vasculature density of neuroblastoma. Their premise is that a better delineation of vasculature density would allow for safer complete resection of viable tumor tissue, hence differentiating macroscopic regions within the tumor of malignant tissue from necrotic and surrounding normal tissue. Changes within the tumor vasculature could be used in-40 intraoperatively as a diagnostic tool to guide surgical resection. They reviewed the mean vascular density of various NB subtypes after induction chemotherapy using scanned histological samples. They also evaluated one patient prospectively enrolled and an ex vivo photoacoustic imaging scan was performed on two representative sections. They found that post-chemotherapy, viable areas of differentiating NBs and ganglioneuroblastomas were associated with higher MVD when compared to poorly differentiated ones. Also, early necrotic regions showed higher MVD than late necrotic and viable regions. Calcified areas showed higher MVD. Intraoperative detection with a high-definition optical imaging device. (PAI). Overall, this is an interesting paper despite the limitations associated with IHC, imaging analyses. And a total number of samples analyzed were somewhat incremental. More importantly, their data demonstrated an association of MVD to neuroblastoma prognosis; the authors should include additional experiments demonstrating cause and effect mechanistic data. Additionally, it isn’t clear how would surgeons use the vascular density of NB information in the operating room. 

Author Response

1. We would like to thank the reviewer for the valuable and detailed feedback. Regarding the number of samples, 5 regions for each of the selected microscopical areas (diagnosis, viable tumor, early necrosis, late necrosis, and calcified tissue) were analyzed for each of the patients included in the study. This led to around 200 regions being manually labeled for each of the histological subtypes included in the study, which gave enough statistical power to prove a significant difference between viable and non-viable areas (primary aim of the paper). As stated in the material sections, the number of regions was established using a Systematic Uniform Random Sampling method, which showed how five regions gave a reliable estimate of the patient’s vascular density in a defined microscopical area. Nonetheless, we agree that a bigger number of samples could have highlighted differences in the microvascular density of viable areas of different histological subtypes. However, as mentioned above, this was not the primary aim of the paper.

2. Indeed, the association between MVD and Neuroblastoma prognosis would be interesting to look at, and although this didn’t reach significance, our results did show a higher microvascular density in unfavorable histology groups compared to more differentiated subtypes. However, this manuscript focuses on the use of differences in vascular density as a preliminary data to use PAI as a novel tool to predict different histology intraoperatively. Therefore, we believe that additional experiments would not fit within the scope of the present study. We thank you for your suggestion, and we will definitely develop a second paper to investigate cause-effect mechanistic data.

3. Thank you for pointing this out. The surgeons will use a PAI scanner during the surgery, and the images of the microvasculature will be used to intraoperatively define viable vs non-viable (necrotic, calcified, or fibrotic) areas of neuroblastoma. This may advance the surgery, reducing the extension of the resection, particularly in areas at high risk of vascular complications or organ damage (i.e. aorta and its branches, kidney vasculature). We have now clarified this important concept in the discussion section of the manuscript:

“Surgery plays a pivotal role in the multimodal treatment of NB [39,40]. At the current state of the art, pediatric oncology surgeons exclusively rely on visual inspection, palpation, and personal experience to identify tumor location and extension. Our results show a distinct microvascular signature between viable and non-viable areas. We believe that this difference, combined with the ability of PAI to provide a non-invasive high-resolution 3D imaging of vascular morphology, will inform surgeons during tumor resection, providing an important innovation in the field. In detail, the presence of live imaging at the time of resection could guide the surgeon to better and selectively confirm the presence of viable tumor in difficult anatomic areas, reducing morbidity and mortality.

While imaging techniques, like magnetic resonance imaging and computed tomography, often necessitate the use of an exogenous contrast agent to achieve high-resolution images of the vasculature, PAI relies on the strong light absorption of hemoglobin which enables the excellent definition of the microvasculature without employing exogenous contrast [41]. Previous studies have focused on the applicability of PAI for the evaluation of the vascular morphology of solid tumors to either assess pharmacodynamic response to drug treatment or detect cancer based on intrinsic optical absorption contrast [41, 42]. Our pilot data also confirm the possibility for PAI to clearly demarcate nodules of viable tumor and distinguish regions with different responses to chemotherapy. Future work will focus on the ex-vivo acquisition of high-resolution 3D anatomical images of tumor vasculature, both in formalin-fixed and fresh tissue, using our preclinical Fabry-Perot PAI system. Ex-vivo validation of PAI’s ability to distinguish viable versus non-viable tissue, with the use of histopathology as a gold standard, will be of crucial importance before translating this novel technology into the clinical environment.”

Reviewer 3 Report

This is a manuscript on the changes in microvascular density's ability to predict viable and non-aviable areas in high-risk neuroblastoma. It is a well-written manuscript and besides adding to the literature on angiogenesis in neuroblastoma, there are important messages on how to further improve the understanding of the subject and the management of neuroblastoma.

Neuroblastoma after induction chemotherapy can be heterogeneous in tumor viability. Therefore, the ability to determine its viability may aid in decisions to excise tumors are surgically-risky areas. One has to be cautious in deliberately leaving "non-viable" tumor in-situ as we have yet determined if these will lead to future recurrence. Nevertheless, this manuscript will most likely lead to more interesting discussions and research. 

Author Response

Thank you for your kind and valuable feedback. We completely agree with you. The data presented in the paper are still preliminary, and they will require strong validation before clinical adoption.  The aim of this paper was to confirm the possibility of using vascular density to distinguish between viable and non-viable tumor regions and to prove that PAI can visualize these differences in ex-vivo human samples. We are planning to perform a large prospective study to validate the use of PAI on ex-vivo human samples, and we will use histopathology confirmation to assess sensitivity and specificity. This has now been clarified in the discussion section:  

“Our pilot data also confirm the possibility for PAI to clearly demarcate nodules of viable tumor and distinguish regions with different responses to chemotherapy. Future work will focus on the ex-vivo acquisition of high-resolution 3D anatomical images of tumor vasculature, both in formalin-fixed and fresh tissue, using our preclinical Fabry-Perot PAI system. Ex-vivo validation of PAI’s ability to distinguish viable versus non-viable tissue, with the use of histopathology as a gold standard, will be of crucial importance before translating this novel technology into the clinical environment.”
